# ROYAL SOCIETY
# OPEN SCIENCE

mathematical modelling/health and disease and epidemiology/differential equations

COVID-19, convalescent blood transfusion, logistic feasibility, reduction of deaths, mathematical models

**Author for correspondence:**
Jianhong Wu
e-mail: wujh@yorku.ca

†Contributed equally as the first author to this study.

# Effectiveness and feasibility of convalescent blood transfusion to reduce COVID-19 fatality ratio

Xi Huo[1], Xiaodan Sun[2,†], Nicola Bragazzi[3] and Jianhong Wu[3]

[1]Department of Mathematics, University of Miami, Coral Gables, FL 33146, USA
[2]School of Mathematics and Statistics, Xi'an Jiaotong University, Xi'an 710049, Shaanxi, People's Republic of China
[3]Laboratory for Industrial and Applied Mathematics, Department of Mathematics and Statistics, York University, Toronto, Ontario, Canada M3J 1P3

(iD) XH, 0000-0001-6308-5118; JW, 0000-0003-0052-5336

*Background:* As of December 2020, COVID-19 has spread all over the world with more than 81 million cases and more than 1.8 million deaths. The rapidly increasing number of patients mandates the consideration of potential treatments for patients under severe and critical conditions. Convalescent plasma (CP) treatment refers to the approach of infusing patients with plasma from recently recovered patients. CP appears to be a possible therapeutic option to manage patients suffering from severe or even lethal infectious disorders, in which 'traditional therapies' have failed to obtain any result. *Methods:* In the present study, we develop a mathematical model on the treatment-donation-stockpile dynamics for an optimal implementation of CP therapy to examine potential benefits and complications in the logistic realization of this therapy in a large-scale population. We parametrize the model with COVID-19 epidemics in Italy, and conduct scenario analyses to estimate outcomes of population-wide CP therapy and to examine the maximum number of CP donation processions per day. *Results:* Under the assumption that the efficacy of CP is 90%, we show that by the end of year 2020, initiating the population-wide CP therapy from April 2020 can save as many as 19 215 lives (ranging from 5000 to 28 000 depending on donor availability), while the demand for apheresis use is manageable in all scenarios: the maximum daily demand is 156 (ranging from 27 to 519 depending on donor availability) for the first outbreak wave and 1434 (ranging from 224 to 4817 depending on donor availability) for the second wave. Given that Italy has 61 centres with apheresis this maximum demand level corresponds to a daily average of 2.5 and 23.5 processions

of CP donation being performed by each centre with respect to each outbreak wave. *Conclusions:* Our analyses show that population-wide CP therapy can contribute to curbing COVID-19-related deaths, and the logistic implementation is feasible for developed countries. The reduction of deaths can be very significant if the CP therapy is started earlier in the outbreak, and remains significant even if it is implemented during the outbreak peak time.

# 1. Background

On 11 March 2020, the World Health Organization (WHO) declared the currently ongoing COVID-19 outbreak as a pandemic. As of December 2020, COVID-19 has spread to 223 countries with more than 81 million cases and more than 1.8 million deaths. While it remains a global challenge to identify epidemiological determinants of the novel coronavirus, termed as SARS-CoV-2 (previously known as 2019-nCoV), and to develop and implement effective surveillance, containment and mitigation strategies, the rapidly increasing number of patients calls for effective treatments for patients under severe and critical conditions.

Convalescent plasma (CP) treatment, also known as passive immune therapy, refers to the approach of infusing patients with plasma from recently recovered patients. CP appears to be a possible therapeutic option to manage patients suffering from severe or even lethal infectious disorders, in which 'traditional therapies' have failed to obtain any result. More specifically, patients are administered plasma, fractioned antibodies or other specific blood products from convalescent donors, thus receiving immunoglobulins and other purified molecules and compounds that should neutralize the virus and reduce the viral load. Such a treatment is theoretically promising, as critically ill patients receive healing factors from those who have survived the disease, which will potentially save their lives [1]. This should lead patients to recovery: however, the precise mechanisms underlying plasma therapy are not very well understood yet.

Plasma therapy was first empirically used during the 1917–1918 Spanish flu pandemic before the development and introduction of antibiotics, and, since then, has been employed during various epidemic/pandemic outbreaks, including the 2003 SARS-CoV-1, the 2009–2010 H1N1 influenza virus, the 2012 MERS-CoV, the 2013–2015 Ebola virus and the 2015 avian-origin influenza A (H7N9) outbreaks [1,2]. In particular, during the 2014–2015 Ebola outbreak in West Africa, the WHO issued an interim guidance to the use of CP and whole blood in large-scale treatment of the infection [3]. On 24 March 2020, the US Food and Drug Administration (FDA) allowed healthcare providers to administer CP as an investigational treatment for COVID-19. From 3 April to 2 June 2020, the FDA Expanded Access Program for COVID-19 Convalescent Plasma transfused a convenience sample of 20 000 hospitalized patients with COVID-19 convalescent plasma, and demonstrated the safety of this therapy [4].

However, the effectiveness of CP treatment has not been fully demonstrated yet. Furthermore, such a treatment is particularly challenging in terms of the logistical feasibility of population-wide administrations. On the other hand, given the current absence of effective pharmacological options and in consideration of the global spread of COVID-19 [2], the effectiveness of CP and the anticipated challenge for its large-scale use in the population needs to be addressed through pilot studies. Computational/mathematical simulations represent a tool to address this critical issue of effectiveness and feasibility.

The implementation of a large-scale passive immunotherapy programme requires the coordination of donor screening and selection, plasma collection and stockpiling, as well as treatment authorization and delivery. These aspects have been investigated only by a few mathematical models. For instance, the logistical feasibility and potential benefits of using CP to treat severe H1N1 cases in Hong Kong was assessed, where 0.5% of infected cases were severe [5]. Possible reductions in the case fatality ratio via a hypothetical CP therapy in the population were estimated for the 2014–2015 Ebola outbreak in West Africa, where all cases were severe [6].

In terms of the COVID-19 outbreak, current knowledge shows that 81% of all cases are mild, 14% are severe and 5% are critically ill, with most of the deaths occurring in the last group [7]. In the present study, we develop a mathematical model with treatment-donation-stockpile dynamics for the optimal implementation of CP therapy to examine potential benefits and complication in the logistic realization of this therapy in a large-scale population. The modelling framework is suitable for any population; however, we consider Italy as a case study. As both of the previous studies

aforementioned [5,6] emphasized that the demand–supply balance is driven by the actual disease dynamics, we first calibrate the treatment dynamics from the reported case data, so as to simulate the dynamics of convalescent and critically ill patients in both outbreak waves in Italy. We then address the following questions: (i) reduction in the overall deceased cases with various assumed efficacy of the CP therapy; (ii) peak demand of hospital beds, treatment facilities and plasma collection centres under various scales of the CP implementations; and (iii) impacts of timings for the start of the population-wide CP therapy.

# 2. Methods

Our goal is to simulate the assumed process of treating patients with severe symptoms by plasma collected from convalescent patients with mild symptoms. We start with a baseline compartmental model to simulate the patient dynamics without the implementation of CP therapy, and parametrize this baseline model with respect to the real data. This will help inform the daily number of critical cases, daily recovered individuals and potential donors. We will then develop the treatment-donation-stockpile model for a complete simulation of the large-scale use of CP therapy.

## 2.1. Baseline model

The population of all identified patients is stratified into four compartments: patients with mild symptoms, critically ill patients in intensive care units, recovered individuals and deceased patients. The model structure is shown in figure 1, and equations are shown in electronic supplementary material, *Appendix*.

The inflow rate of newly identified cases, denoted as $\Lambda(t)$, is solely driven by the disease dynamics and can be calibrated from daily reported case data. We assume that all patients would initially have mild symptoms upon identification, and some of them will develop critical conditions at a rate $p$. Patients with mild symptoms will recover after an average of 8 days, and we assume no deaths for this patient group [7]. For patients with critical conditions, we assume they will either recover within an average of 10 days [8] or will die at rate of $\mu_P^c$.

## 2.2. Data collection and fitting

We fitted the two outbreak waves separately. For the first outbreak, we used data collected from 23 February to 2 June 2020 about Italy's daily reports on daily reported cases, cumulative deaths, and daily number of cases in intensive care units (ICUs) [9]. For the second outbreak, we made use of the same type of data from 20 September 2020 to 11 January 2021. We calculated the 7-day moving average of the daily reported cases to obtain $\Lambda(t)$, so as to reduce the inflow rate fluctuations caused by the variations of testing availability.

The deterioration rate $p$ and death rate $\mu_P^c$ were fitted correspondingly to each wave via the least-square fitting method. That is, we estimated the parameters by minimizing the sum of square error between the reported data and model prediction values

$$\sum_{\text{day}\,i} (D_i - \hat{D}_i)^2 + (\text{ICU}_i - \hat{\text{ICU}}_i)^2,$$

where $D_i$ and $\text{ICU}_i$, respectively, denote the cumulative deaths and number of patients in ICUs predicted by the model for day $i$, where $\hat{D}_i$ and $\hat{\text{ICU}}_i$ denote the corresponding values from data. The fitted parameter values are listed in table 1 and fitting outcomes are summarized in electronic supplementary material, *Appendix*.

In the simulation below, the inflow rate of new patients $\Lambda(t)$ is calibrated as stated above from the reported data from 23 February to 31 December 2020. We use the values of $p$ and $\mu_P^c$ estimated from the first wave for the time period from 23 February to 20 September 2020 and those estimated from the second wave for the time period after 20 September 2020 till the end of year 2020.

## 2.3. Treatment-donation-stockpile dynamics

The population-wide use of CP therapy would involve treatment approval, donor management, and CP stockpile and supply. Based on the guidelines of applications for CP therapy as investigational treatment

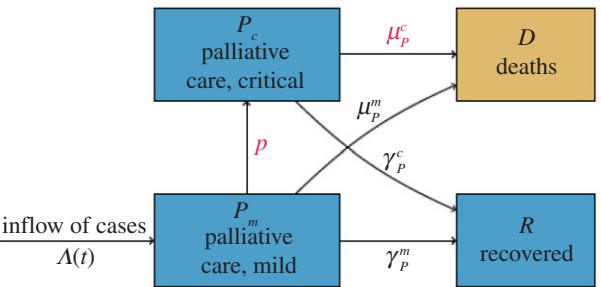

**Figure 1.** Compartmental diagram for the baseline treatment dynamics model. $\Lambda(t)$ records the daily identified cases calibrated from real data, $p$ and $\mu_P^\varsigma$ as coloured in red are the parameters fitted from the ICU and mortality data.

provided by FDA [2], we assume only critically ill patients are considered for CP therapy and the availability of the therapy is highly dependent on the total storage of CP at the time. Further, donations are processed in centres with apheresis facilities and there are 61 such centres in Italy [10], and our model will simulate the demanded workload with apheresis. The compartmental dynamics in terms of the population and CP stockpile are illustrated in figure 2. Table 1 provides all parameters involved in our simulation.

*Stockpile compartment:* we assume all donated CPs are stored and prepared for distribution in facilities serving as blood banks, and denote $B(t)$ as the total plasma bank storage at time $t$. CPs would expire on an average of 40 days according to the WHO guidelines [3]. Each patient will get a single dose of plasma, thus the consumption rate of CP is equivalent to the rate of patients receiving CP therapy which is illustrated below.

*Treatment compartments:* we further stratify critically ill patients into two classes: those under palliative care or other treatments, and those who are treated with CP. Newly identified individuals are assumed to have mild symptoms and be under palliative care ($P_m$); such patients would develop critical symptoms at the rate $p$ and be treated via palliative care or other methods ($P_c$). Then, the critically ill patients will receive CP therapy at a rate that depends on a saturation function about the current blood bank storage, $\eta(B/(B+K))$, where $\eta$ is the arrangement and approval rate for blood transfusion therapy, and the term $B/(B+K)$ models the fraction of patients that could be offered CP therapy—which depends monotonically on the plasma stockpile. Specifically, $B$ is the daily blood bank storage, and $K$ represents the threshold of CP level where only $1/2$ of the demands can be met (thus higher values of $K$ refers to more conservative use of the stockpile). Patients under palliative care would either recover or die at rates illustrated in the baseline model. Currently there is limited knowledge of the efficacy of CP therapy based on large-scale population data [11], we assume it to be $f$ and will discuss outcomes based on different assumptions.

*Donation compartments:* we assume only patients survived from mild symptoms will enter the recovered class $R$ (the pool for potential donors), and will be ready for donor screening within an average of 14 days [2,3]. We assume a percentage $\varepsilon$ of the convalescent patients would be eligible to donate their CPs and denote such population as $D_1$. Eligible donors would make donations at a rate $\alpha$ and soon move to class $D_0$ for rest, meanwhile their CP donations are transferred to the blood bank. According to the WHO interim guidance, donors at rest will need to wait for 14 days to become eligible for subsequent donations, and we categorize those who are qualified for multiple time donations as $D_m$. Similarly, donors in $D_m$ would make donations at the rate $\alpha$ then soon move back to the donor at rest class. Donors would quit the process at a rate of $\xi$.

To better parametrize the donation rate $\alpha$ and quitting rate $\xi$, we relate them to the parameters $p_d$ and $p_c$, respectively, where $p_d$ represents the possibility for a newly qualified donor to make a donation on his/her first day and $p_c$ refers to the probability for a newly qualified donor or a donor who has just finished his/her last donation to quit the donor system. The relationship between the rate $\alpha$ and the fraction $p_d$ can be derived from an easy exercise: suppose we keep track of a group of qualified donors $x(t)$ at time $t$, and assume all of them will eventually make donations; we have $x'(t) = -\alpha x(t)$. Then, the fraction of donors who would complete their donations by the first day is $1 - e^{-\alpha}$, and we can thus relate $p_d = 1 - e^{-\alpha}$. A similar derivation yields $p_c = 1 - e^{-\xi}$. In short, larger values of $p_d$ and smaller values of $p_c$ represent better adherence and stronger willingness for donations of the majority convalescent patients. Further, we assume recovered patients with mild symptoms are lost at the rate of $\xi$ before the implementation of CP therapy thus would not be traced for donation.

**Table 1.** Table of parameters.

| parameter | definition | values | reference |
|---|---|---|---|
| transmission parameters | | | |
| $p$ | progression rate from mild to critical | 0.03653 d$^{-1}$ (1st wave) | Fitted. |
| | | 0.00477 d$^{-1}$ (2nd wave) | |
| $\gamma_P^c$ | recovery rate of critical case | 0.1 d$^{-1}$ | Assumed [8]. |
| $\gamma_P^m$ | recovery rate of mild case | 0.125 d$^{-1}$ | Assumed [8]. |
| $\mu_P^c$ | death rate of critical case | 0.2423 d$^{-1}$ (1st wave) | Fitted. |
| | | 0.1692 d$^{-1}$ (2nd wave) | |
| $\mu_P^m$ | death rate of mild case | 0 | Assumed [7]. |
| treatment parameters | | | |
| $\gamma_T$ | recovery rate with CP | 0.0256 d$^{-1}$ | Derived. |
| $\mu_T$ | death rate with CP | $\frac{f}{1-f}\gamma_T$ | Derived. |
| $\eta$ | arrangement rate of CP therapy | 1 d$^{-1}$ | Assumed. |
| $K$ | threshold of blood stock | varied ([a]10 000) | — |
| $f$ | efficacy of CP therapy | varied ([a]90%) | — |
| donation parameters | | | |
| $\varepsilon$ | probability of becoming a donor | varied ([a]0.1) | — |
| $\sigma$ | transition rate from discharged patients to potential donors | 1/14 day$^{-1}$ | Assumed. |
| $\alpha$ | rate of donation | $-\ln(1 - p_d)$ | Derived. |
| $\omega$ | inflow rate of donation to blood bank | $\alpha$ | [6] |
| $p_d$ | probability of making donation | varied ([a]0.1) | — |
| $\xi$ | loss rate of donors | $-\ln(1 - p_c)$ | Derived. |
| $p_c$ | probability of quitting donation | varied ([a]0.1) | — |
| $\tau$ | donor recovery period between consecutive donations | 14 days | [3] |
| $\lambda$ | CP expiration rate | 1/40 d$^{-1}$ | [3] |
| initial values on 23 February 2020 | | | |
| $P_c(0)$ | critically ill cases | 26 | ICU data. |
| $P_m(0)$ | mild cases | 76 | Data. |
| $R(0)$ | recovered population | 1 | Data. |
| initial values on the first day of CP implementation | | | |
| $T(0)$ | patients under CP treatment | 0 | Assumed. |
| $D_1(0)$ | potential first-time donors | 0 | Assumed. |
| $D_0(0)$ | donors under recovery | 0 | Assumed. |
| $D_m(0)$ | potential multi-time donors | 0 | Assumed. |
| $B(0)$ | blood bank stockpile | 0 | Assumed. |

[a]The baseline value of the parameter.

# 3. Results

There are several categories of parameters related to different aspects of CP therapy: donor related ($\varepsilon$, $p_c$, $p_d$); stockpile usage related ($K$); CP therapy efficacy related ($f$); and importantly, the beginning date of the population-wide CP collection and treatment. Clearly, all donor-related parameters and the timing of population-wide treatment will have impact on the number of overall avoided deaths and the implementation burden (both donor screening and donation collection require specific personnel and facilities [5]), while $K$ and $f$ have impact only on the number of avoided deaths. Our simulations are designed to quantify these impacts.

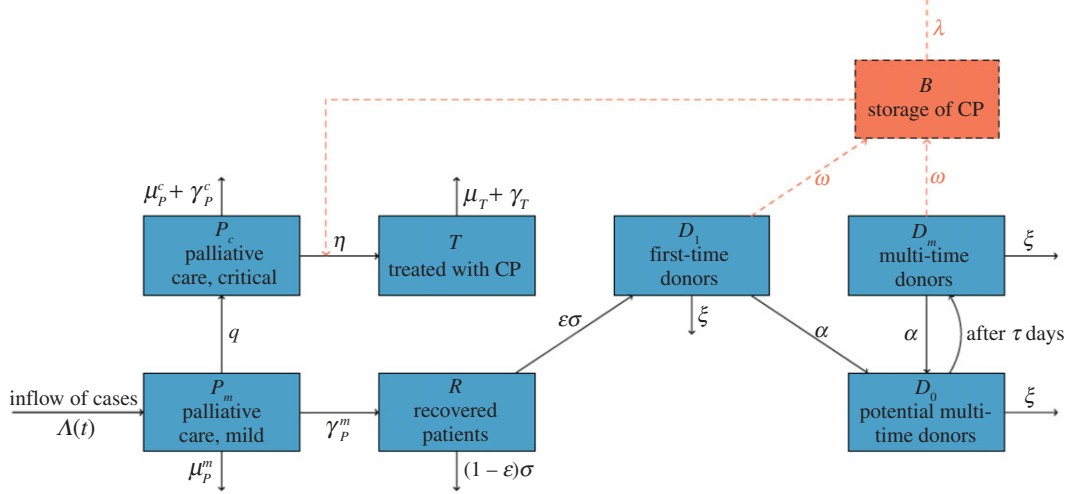

**Figure 2.** Compartmental model for the treatment-donation-stockpile dynamics of CP therapy. Blue compartments represent the population dynamics of patients and convalescent individuals, red arrows and compartment refer to the dynamics of blood products.

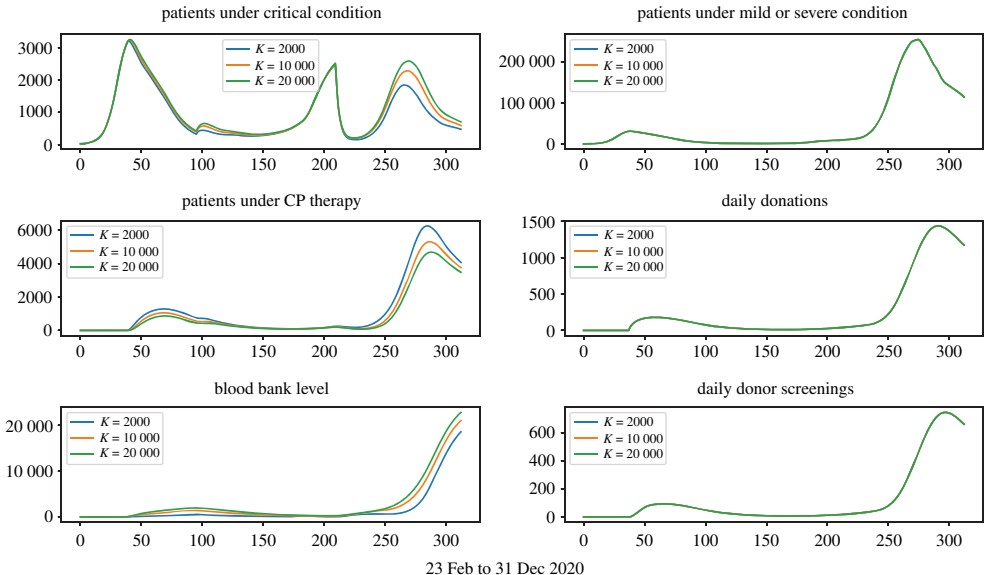

**Figure 3.** Treatment-donation-stockpile dynamics under different conservation strategies of CP stockpiles. Larger $K$-values correspond to more conservative deployment of CP stockpiles.

Figures 3–6 show that under wide ranges of all parameters: (Dscr) the daily demand for donor screening (i.e. the process to evaluate whether or not a convalescent patient is a qualified donor) is not much influenced in the first wave due to the relatively small scale of the outbreak, whereas the screening demand would be affected significantly by donor availability. The peak time demand for donor screening will fall below 250 convalescent patients per day for the first outbreak and below 1500 for the second wave; (CPCol) the daily demand for CP collections (i.e. the daily number of donations needing be processed) is sensitive to all donor-related parameters; (CPPat) the daily number of patients under CP therapy is sensitive to all parameters investigated, as well as the daily blood bank stockpile level. Two observations are important:

  (i) Figure 7 shows that early start of population-wide CP treatment will enable more CP treatments, and the daily demands for apheresis (i.e. number of donations to be processed) will be maintained at a reasonable level.

 (ii) Figure 5 shows that a lower $p_d$ value would result in a higher number of daily donations, this is because $p_d$ measures the possibility for donors to make donation on the first date of their

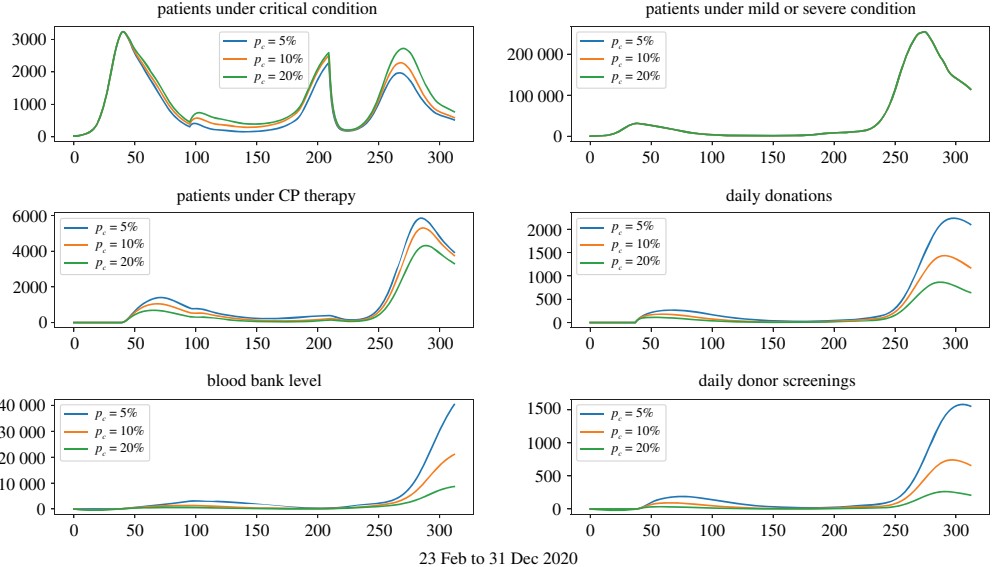

**Figure 4.** Treatment-donation-stockpile dynamics under different loss-of-donor probabilities. Larger $p_c$ values correspond to more daily loss of qualified donors.

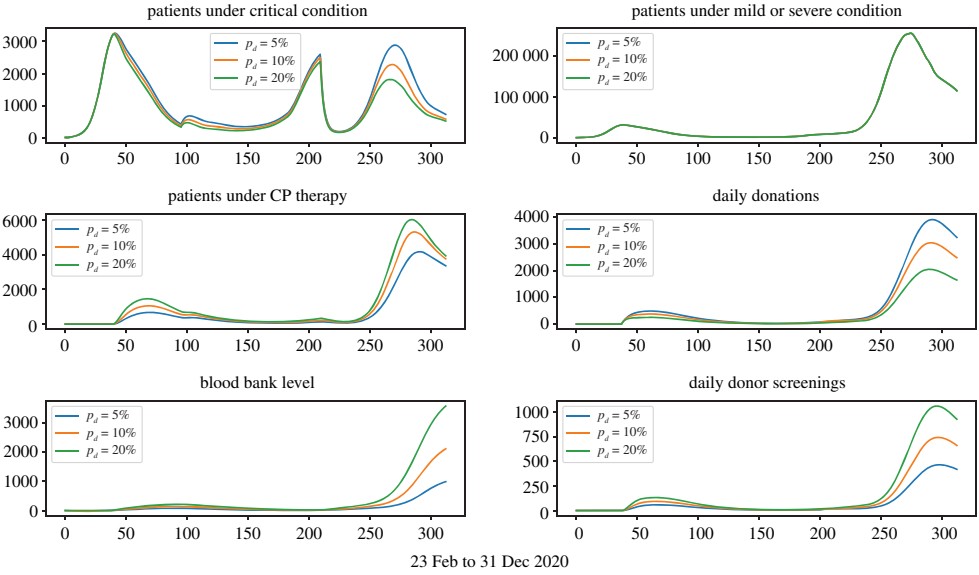

**Figure 5.** Treatment-donation-stockpile dynamics under different daily donation probabilities. Larger $p_d$ values correspond to higher daily donation rates.

qualification, so even if a small $p_d$ value refers to a lower possibility of donation on the first date, it also refers to more donors ready for donation in the future.

Figure 8 describes the impact of donor-related parameters and timing of the start of CP treatment on the reduced number of deaths by the end of year 2020 (CP impact on mortality) and peak time demand (CP-logistic feasibility) for apheresis (under the assumption of a 90% CP efficacy):

(iii) Enhancing the donor adherence (i.e. the donor-related parameters) and early implementation of large-scale CP therapy can save thousands of lives.

(iv) The donor adherence variation will significantly affect the burden of apheresis, while an early start will not.

(v) The transmission receded during the summer months, thus initiating the CP therapy during this time would not significantly reduce the number of fatal cases; however, in reality early implementation would lead to better coordinated logistical arrangements thus help to prepare for the second outbreak.

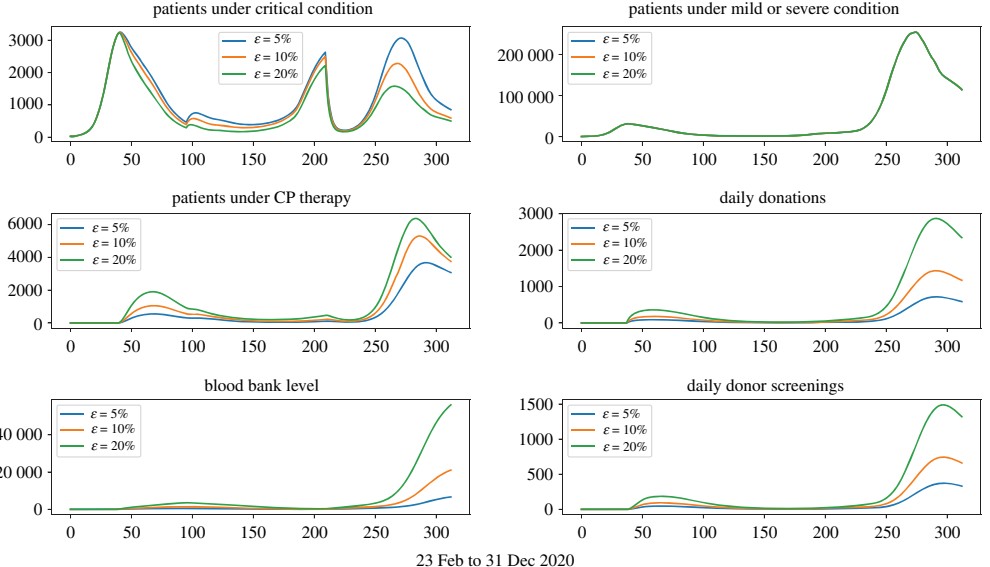

**Figure 6.** Treatment-donation-stockpile dynamics under different ratios of qualified donors.

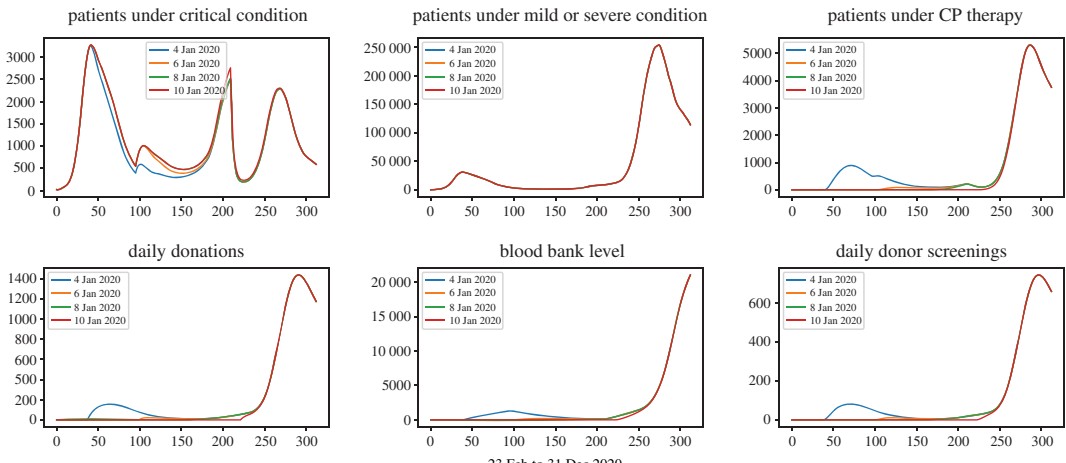

**Figure 7.** Treatment-donation-stockpile dynamics under different timings to start the population-wide CP therapy.

Figure 9 shows that the actual efficacy of CP therapy is vital in curbing the total number of deaths: in order to achieve the best outcome, the efficacy of CP therapy should be above 80% and the population-wide collection and treatment should start by early April, in the Italy scenario we are simulating.

Figure 10 compares the peak time apheresis demand for both waves under different start time of the CP collection. From the simulation, we observe that the maximal demand for apheresis would be affected by the initiation time of CP collection only when the implementation starts in the middle of an outbreak wave. Further, the demand for apheresis is manageable on all occasions—the maximum demand is 1434 per day, given 61 centres with apheresis this corresponds to a daily average of 23.5 processions of CP donation being performed by each centre.

# 4. Conclusion

By the time Italy hit its second wave of COVID-19 outbreak, there were no drugs to effectively treat COVID-19 patients and there were no approved vaccine products. By the end of year 2020, vaccines have not been distributed to the majority of population. Given that no therapeutic can properly

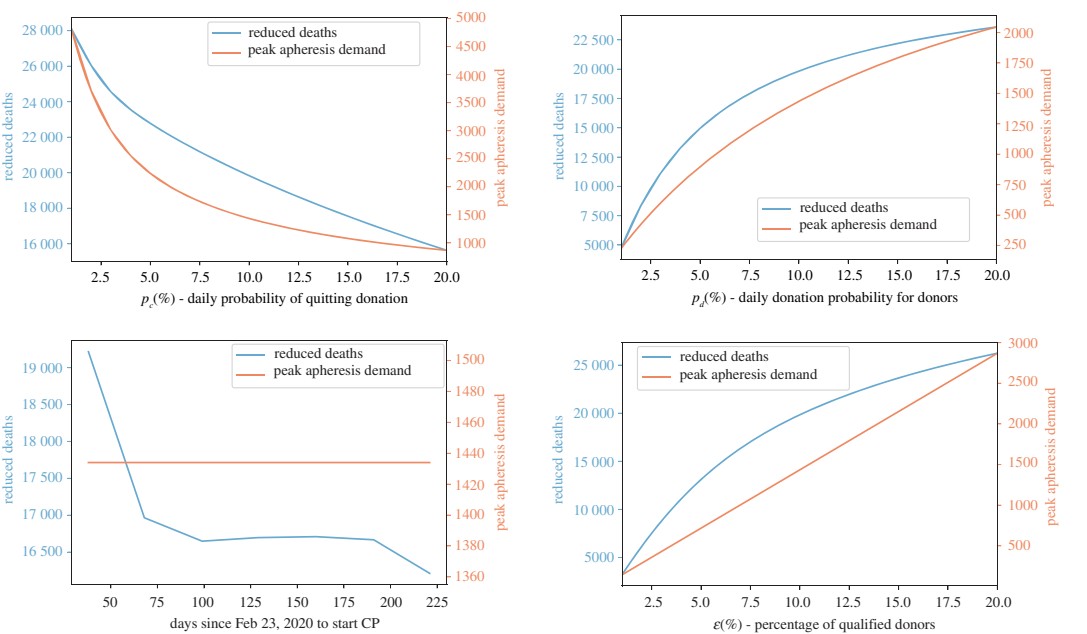

**Figure 8.** Reduced deaths versus apheresis demands. The left-hand side axis of each figure measures the total number of deaths that could be avoided by the end of year 2020 via applying CP therapy under various levels of donation adherence ($p_c$), donation participation ($p_d$), start time and percentage of qualified donors ($\varepsilon$). The right-hand side axis of each figure measures the peak time demand of apheresis.

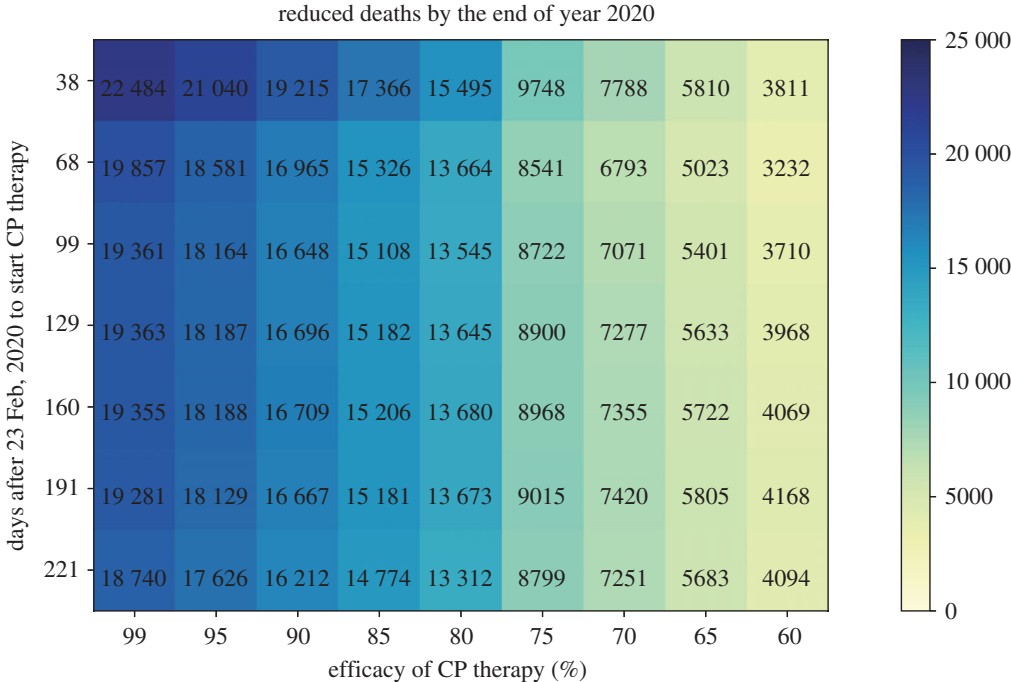

**Figure 9.** Reduced deaths by the end of year 2020. We calculate the number of deaths avoided by the implementation of CP therapy under hypothesized CP therapy efficacy and timings to start the population-wide CP collection.

counteract and mitigate the COVID-19 burden, besides non-pharmacological interventions, including self-isolation, quarantine and lockdown of territories/communities, CP remains an empirical treatment worthy of being explored.

Our analyses, based on the treatment-donation-stockpile dynamics, show that a population-wide CP treatment can contribute to curbing COVID-19-related deaths. The reduction of COVID-19 deaths can be very significant if the CP therapy is started earlier in the outbreak, and remains significant even if it is

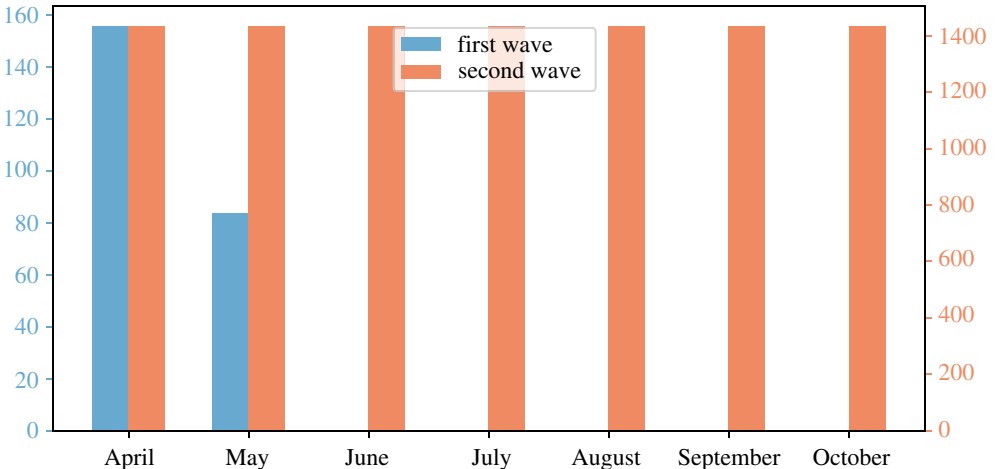

**Figure 10.** Peak time apheresis demand with respect to starting month. Each orange bar represents the peak time demand of apheresis during the second outbreak wave given the large-scale CP treatment and collection started at the beginning of each month in 2020. The blue bars represent the demand during the first outbreak.

implemented after the outbreak peak time. The logistic implementation is feasible, using the Italy epidemic as example, simulations under the baseline parameter values showed the maximum demand of apheresisis 1434 per day corresponds to a daily average of 23.5 processions of CP donation being performed by each centre, given that the country has 61 centres with such a capacity. Our model-based simulations also show that qualified donors should be encouraged to make donations as early as possible if we want to have a more operational system for CP treatment, blood donation and stockpiling. Our model is portable and can be adopted in other settings to address both the impact of COVID-19 mortality reduction and the logistic operation feasibility.

## 5. Discussion

We note that little is known about the effectiveness of CP against COVID-19, so we have to refer to the existing scholarly literature and studies assessing CP in the treatment of pandemic outbreaks or particularly severe and life-threatening infectious diseases. Concerning the 1917–1918 Spanish flu pandemics [12], a recent meta-analysis has pooled together eight studies, totalling a sample of 1703 patients suffering from severe influenza pneumonia and treated with convalescent plasma versus untreated controls managed in the same hospital/ward. Authors computed an overall crude case-fatality rate of 16% and 37% among treated and untreated patients, respectively, with an absolute risk difference in mortality between the two groups of 21% [95%CI 15–27%], ranging from 8% to 26%. The major determinant of case-fatality rate was the timing of the administration of blood products: if received earlier the mortality rate was 19% and increased up to 59% if received later, with an absolute risk difference in mortality of 41% [95%CI 29–54%], ranging from 26% to 50%. This is in line with the findings based on our mathematical model. Furthermore, the rate of adverse effects was low and reactions were generally mild, including chill reactions and exacerbations of symptoms. Generalization of the findings was, however, hindered by some methodological flaws affecting the included studies, with the lack of blinded, randomized, placebo-controlled trials being the major shortcoming.

Similar limitations affect the trials of convalescent plasma used for treating Ebola and avian-origin influenza A viruses. Concerning COVID-19, few studies exist. For instance, Duan and colleagues [11] recruited 10 critically ill adults and found that one dose (200 ml) of CP, besides being well tolerated and without side-effects, could significantly elicit the formation of a high amount of neutralizing antibodies, counteracting COVID-19-related viraemia and radiological manifestations in a week, with an improvement of clinical symptoms in 3 days. Similarly, Zhang and co-workers [13] recruited four patients suffering from a severe form of infection (including a pregnant woman), who quickly recovered after CP administration. In addition, Shen and coauthors [14] treated with CP a series of five patients aged 36–65 years (two women) and observed a significant improvement in clinical and radiological symptoms as well as in viraemia. However, despite such promising findings emerging

from existing medical case reports and case series as well as from the present mathematical model, high-quality randomized trials should be conducted.

Data accessibility. Data and relevant code for this research work are stored in GitHub: https://github.com/xxh314/COVID19-bloodtransfusionmodel and have been archived within the Zenodo repository (doi:10.5281/zenodo.4315455).

Authors' contributions. X.H., X.S. and J.W. developed the model; N.B. collected data and medical references; X.H. and X.S. performed data fitting and numerical simulations; all authors analysed the results and contributed in writing the manuscript. All authors read and approved the final manuscript.

Competing interests. We declare we have no competing interests.

Funding. This research was partially supported by the National Science Foundation (grant no. DMS-1853622 (X.H.)) and Direct and Indirect Effects of COVID-19 Program from College of Arts and Sciences, University of Miami (X.H.); by the National Natural Science Foundation of China (grant no. 11701442 (X.S.)), the Fundamental Research Funds for the Central Universities (grant no. xzy032020028 (X.S.)), the Fundamental Research Funds for the Central Universities (grant no. xjj2018259 (X.S.)); by the Canada Research Chair Program (grant no. 230720 (J.W.)), the Natural Sciences and Engineering Research Council of Canada (grant no. 105588-2011 (J.W.)), and the Canadian Institute of Health Research (CIHR) 2019 Novel Coronavirus (COVID-19) rapid research program (N.B. and J.W.).

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
