## [Peer Review File · Royal Society Open Science]

Review History

RSOS-202248.R0 (Original submission)

Review form: Reviewer 1

Is the manuscript scientifically sound in its present form?

Yes

Are the interpretations and conclusions justified by the results?

Yes

Is the language acceptable?

Yes

Do you have any ethical concerns with this paper?

No

Have you any concerns about statistical analyses in this paper?

No

Recommendation?

Accept with minor revision (please list in comments)

Comments to the Author(s)

Please see attached files (Appendix A).

Review form: Reviewer 2 (David Gurarie)**Is the manuscript scientifically sound in its present form?**

No

Are the interpretations and conclusions justified by the results?

Yes

Is the language acceptable?

Yes

Do you have any ethical concerns with this paper?

No

Have you any concerns about statistical analyses in this paper?

Yes

Recommendation?

Major revision is needed (please make suggestions in comments)

Comments to the Author(s)

See attached review (Appendix B).

Decision letter (RSOS-202248.R0)

Dear Dr Huo

On behalf of the Editors, we are pleased to inform you that your Manuscript RSOS-202248 "Effectiveness and Feasibility of Convalescent Blood Transfusion to Reduce COVID-19 Fatality Ratio" has been accepted for publication in Royal Society Open Science subject to minor revision in accordance with the referees' reports. Please find the referees' comments along with any feedback from the Editors below my signature.

Please submit your revised manuscript and required files (see below) no later than 7 days from today's (ie 11-Jan-2021) date. Note: the ScholarOne system will 'lock' if submission of the revision is attempted 7 or more days after the deadline. If you do not think you will be able to meet this deadline please contact the editorial office immediately.

on behalf of Dr Shigui Ruan (Associate Editor) and Glenn Webb (Subject Editor)
openscience@royalsociety.org

Associate Editor Comments to Author (Dr Shigui Ruan):

Associate Editor: 1

Comments to the Author:

Please revise your manuscript by following the reviewers' comments and address them item by item.

Reviewer comments to Author:

Reviewer: 1

Comments to the Author(s)

Please see attached files.

Reviewer: 2

Comments to the Author(s)

See attached review

===PREPARING YOUR MANUSCRIPT===

Please ensure that you include an acknowledgements' section before your reference list/bibliography. This should acknowledge anyone who assisted with your work, but does not

qualify as an author per the guidelines at <https://royalsociety.org/journals/ethics-policies/openness/>.

===PREPARING YOUR REVISION IN SCHOLARONE===

- Ensure that your data access statement meets the requirements at <https://royalsociety.org/journals/authors/author-guidelines/#data>. You should ensure that you cite the dataset in your reference list. If you have deposited data etc in the Dryad repository, please only include the 'For publication' link at this stage. You should remove the 'For review' link.
- If you are requesting an article processing charge waiver, you must select the relevant waiver option (if requesting a discretionary waiver, the form should have been uploaded at Step 3 'File upload' above).
- If you have uploaded ESM files, please ensure you follow the guidance at <https://royalsociety.org/journals/authors/author-guidelines/#supplementary-material> to include a suitable title and informative caption. An example of appropriate titling and captioning may be found at https://figshare.com/articles/Table_S2_from_Is_there_a_trade-off_between_peak_performance_and_performance_breadth_across_temperatures_for_aerobic_scops_in_teleost_fishes_/3843624.

Author's Response to Decision Letter for (RSOS-202248.R0)

See Appendix C.

Decision letter (RSOS-202248.R1)

Dear Dr Huo,

It is a pleasure to accept your manuscript entitled "Effectiveness and Feasibility of Convalescent Blood Transfusion to Reduce COVID-19 Fatality Ratio" in its current form for publication in Royal Society Open Science. The comments of the reviewer(s) who reviewed your manuscript are included at the foot of this letter.

COVID-19 rapid publication process:

We are taking steps to expedite the publication of research relevant to the pandemic. If you wish, you can opt to have your paper published as soon as it is ready, rather than waiting for it to be published the scheduled Wednesday.

This means your paper will not be included in the weekly media round-up which the Society sends to journalists ahead of publication. However, it will still appear in the COVID-19 Publishing Collection which journalists will be directed to each week (<https://royalsocietypublishing.org/topic/special-collections/novel-coronavirus-outbreak>).

If you wish to have your paper considered for immediate publication, or to discuss further, please notify openscience_proofs@royalsociety.org and press@royalsociety.org when you respond to this email.

on behalf of Dr Shigui Ruan (Associate Editor) and Glenn Webb (Subject Editor)
openscience@royalsociety.org

Appendix A

COMMENTS ON RSOS-202248

The authors developed a mathematical model that couples the disease transmission dynamics with the treatment-donation-stockpile dynamics. They presented an interesting exploration for an optimal implementation of CP therapy to examine potential benefits and complications in the logistic realization of this therapy in a large-scale population. The model was parameterized with COVID-19 epidemics in Italy and their results indicate that population-wide CP treatment can contribute to curbing COVID-19 related deaths.

The manuscript overall is well presented. The idea behind the work holds relevance and could be of value to public health. Some issues can be addressed to increase the clarity of the manuscript.

- The transmission rate $\beta(t)$ is assumed to decrease throughout the outbreak because of the gradually enhanced public health interventions. However, the transmission rate (contact rate*susceptibility*infectivity), for COVID-19, is much more complicated than other infectious diseases because of human behaviors and public intervention policies. Many studies indicated that it would be more reasonable to think $\beta(t)$ first decrease, then rebound and relax to some level. Please clarify.
- It would be nice to see at least one important fitting outcome in the manuscript for readers.
- Many studies have shown that the asymptomatic/pre-symptomatic cases are the bulk of the transmission of COVID-19, I am wondering whether it is worth to be considered in this study and whether it will largely affect the number of donors and data fitting?
- lines 38: The transmission from susceptible to infectious population.

Appendix B

Review of RSOS-202248

The authors propose a mathematical model that couples SEIR transmission dynamics of Covid-19 with the Treatment-Donation-Stockpile system for convalescent plasma (CP) used as antiviral therapy in severe cases. The goal is to assess the effect of CP-therapy, its efficacy and implementation strategies, on cumulative disease fatality. To calibrate transmission model parameters they used cumulative data (confirmed cases, deaths, and ICU numbers) from Italy.

The topic is timely and important, and the authors made good effort to address this complex issue. Their modeling approach however, raises several questions

Major comments:

1. The SEIR transmission part of the model (Fig.1) drops all possible disease pathways (e.g. asymptomatic resolution $L \rightarrow R$, or undetected cases $I \rightarrow R$), except hospitalization pathway ($I \rightarrow P$) at rate $\kappa = 1/6$. Where does $1/6$ come from? Given much uncertainty about undetected Covid transmission, the least authors should do is allow some range of κ s, and explore its implications.
2. The key assumption on transmission coefficient $\beta(t)$ - exponential decay in time, looks very strange to me. In anything it should follow a different pattern – initial steep drop (due to lockdown), and the follow-up period maintenance or gradual relaxation. The authors could consult the available mobility data. To fit two epidemic waves (Figs. 3-7) they arbitrarily switch $\beta(t)$ in September. This part needs a revision. The projections for the next 100 days (Fig. 2 appendix) is not convincing
3. The Treatment-Donation-Stockpile part of the system (appendix, (2)) looks reasonable except another vexing issue related to consumption /depletion of the plasma bank, taken as a sigmoid (saturation) function of $B(t)$. It needs an explanation or revision. Whatever CV 'delivery' function $\phi(B)$ is reasonable, its effect on the treated host pool P_c is not $\eta \phi(B) P_c$, used in equation system (2), but $\eta \phi(B)$, where η is drug efficacy. The units for $\phi(B)$ should "# delivered doses/time" (assuming each host gets a single dose).
4. Re model calibration, fitted parameters of Table 1 (appendix) need some ranges of uncertainty.

Minor comments:

1. p4 L21: while there are many uncertainties re mechanisms of plasma therapy, it's not likely to accelerate immune development, but probably would slow it down. I would drop this comment.
2. P6 L57: might be better to replace 'fatality ratio of CP' by CP-efficacy (in term of death prevention)
3. P7 L11-12. Rates α (donor commitment) and ξ (donor loss) are not the proposed 'fractions' (pd,pc). If anything, rate ξ is due to 'loss of interest' + 'loss of antibodies', rate $\alpha = 1/\text{duration decision process (to donate)}$

Overall paper makes a good effort to address complexities of Covid-19, but it need these issues, and an additional work.

Appendix C

Reviewer #1

The authors propose a mathematical model that couples SEIR transmission dynamics of Covid-19 with the Treatment-Donation-Stockpile system for convalescent plasma (CP) used as antiviral therapy in severe cases. The goal is assess the effect of CP-therapy, its efficacy and implementation strategies, on cumulative disease fatality. To calibrate transmission model parameters they used cumulative data (confirmed cases, deaths, and ICU numbers) from Italy.

The topic is timely and important, and the authors made good effort to address this complex issue. Their modeling approach however, raises several questions.

Major comments:

- 1. The SEIR transmission part of the model (Fig.1) drops all possible disease pathways (e.g. asymptomatic resolution $L \rightarrow R$, or undetected cases $I \rightarrow R$), except hospitalization pathway ($I \rightarrow P$) at rate s . Where does $1/6$ come from? Given much uncertainty about undetected Covid transmission, the least authors should do is allow some range of s , and explore its implications.*
- 2. The key assumption on transmission coefficient - exponential decay in time, looks very strange to me. In anything it should follow a different pattern – initial steep drop (due to lockdown), and the follow-up period maintenance or gradual relaxation. The authors could consult the available mobility data. To fit two epidemic waves (Figs. 3-7) they arbitrarily switch in September. This part needs a revision. The projections for the next 100 days (Fig. 2 appendix) is not convincing.*

Response to reviewer: We fully agree with the reviewer's comments on the model about the transmission dynamics and concerns on the projections. Projecting case counts of COVID-19 is extremely hard and is highly likely to be proven wrong after very short period of time – which is also not the major point of this study. The major goal of this study is to simulate the treatment dynamics with the implementation of the blood transfusion therapy, which can be fully decoupled from the transmission dynamics. The only ingredients we need from the transmission dynamics are: the inflow rate of newly identified cases, and estimated values of the symptom deterioration rate and the death rate. Therefore, we completely dropped the transmission model in our revised version, and replaced it by a baseline model that only focuses on the treatment dynamics of patients so as to fit the required parameters to the real data. To further avoid the uncertainties caused by projections, we utilized the real case count data by the end of year 2020 as the inflow rate of identified cases, and conducted all simulations to the same time point. We believe these modifications would not only address the concerns raised by the reviewer, but also make our results more robust (as it no longer depends on projections).

- 3. The Treatment-Donation-Stockpile part of the system (appendix, (2)) looks reasonable except another vexing issue related to consumption /depletion of the plasma bank, taken as a sigmoid (saturation) function of $B(t)$. It needs an explanation or revision. Whatever CV 'delivery' function is reasonable, its effect on the treated host pool P_c is not $\eta \phi(B) P_c$, used in equation system (2), but $\eta \phi(B)$, where η is drug efficacy. The units for $\phi(B)$ should “# delivered doses/time” (assuming each host gets a single dose).*

Response to reviewer: We agree with the reviewer that the part about depletion of plasma bank and treatment delivery needs more explanation, and we have revised this part in the new version. Please note that η does not refer to drug efficacy, but the arrangement rate of CP therapy (that is, the rate of finding a match in the plasma bank and the arrangement of plasma delivery to the hospital), thus the unit of η is “per day”. The saturation function $\phi(B)$ represents the fraction of patients who would be approved for

CP therapy based on the plasma bank level – higher stockpile would lead to higher fraction, thus this is a dimensionless term. We assume each host gets one single dose, so $\eta\phi(B)Pc$ (with a unit of “dose per day” and also “patient per day”) models both the rate of plasma bank depletion and the rate of treatment deployment.

4. *Re model calibration, fitted parameters of Table 1 (appendix) need some ranges of uncertainty.*

Response to reviewer: We used the `find_MAP()` function from PyMC3 package in Python for data fitting, which unfortunately only returns point estimations. Due to the time constraint of the revision, we were not able to explore other fitting methods that gives confidence intervals. If time permits, it would be the best to fit the model by using the Monte Carlo Markov Chain method which would provide estimations of parameters in distributions, and this would add ranges of uncertainties to our simulations results as well. We really appreciate this suggestion from the reviewer, and we hope to refine our simulations in future studies.

Minor comments:

1. *p4 L21: while there are many uncertainties re mechanisms of plasma therapy, it's not likely to accelerate immune development, but probably would slow it down. I would drop this comment.*

Response to reviewer: We have dropped this comment as suggested by the reviewer.

2. *P6 L57: might be better to replace 'fatality ratio of CP' by CP-efficacy (in term of death prevention)*

Response to reviewer: This is a very good suggestion. We have revised this parameter throughout the paper.

3. *P7 L11-12. Rates (donor commitment) and (donor loss) are not the proposed 'fractions' (pd,pc). If anything, rate is due to 'loss of interest' + 'loss of antibodies', rate = 1/" duration decision process (to donate"*

Response to reviewer: We have realized that the original paragraph about the parameters had caused some misunderstandings, and we have revised and elaborated the corresponding part in the new version.

Overall paper makes a good effort to address complexities of Covid-19, but it need these issues, and an additional work.

Response to reviewer: We would like to thank the reviewer for providing so many helpful comments.

Reviewer #2

The authors developed a mathematical model that couples the disease transmission dynamics with the treatment-donation-stockpile dynamics. They presented an interesting exploration for an optimal implementation of CP therapy to examine potential benefits and complications in the logistic realization of this therapy in a large-scale population. The model was parameterized with COVID-19 epidemics in Italy and their results indicate that population-wide CP treatment can contribute to curbing COVID-19 related deaths.

The manuscript overall is well presented. The idea behind the work holds relevance and could be of value to public health. Some issues can be addressed to increase the clarity of the manuscript.

- 1. The transmission rate $\beta(t)$ is assumed to decrease throughout the outbreak because of the gradually enhanced public health interventions. However, the transmission rate (contact rate*susceptibility*infectivity), for COVID-19, is much more complicated than other infectious diseases because of human behaviors and public intervention policies. Many studies indicated that it would be more reasonable to think $\beta(t)$ first decrease, then rebound and relax to some level. Please clarify.*
- 2. It would be nice to see at least one important fitting outcome in the manuscript for readers.*

Response to reviewer: We have provided the fitting outcomes and figures in the Appendix file.

- 3. Many studies have shown that the asymptomatic/pre-symptomatic cases are the bulk of the transmission of COVID-19, I am wondering whether it is worth to be considered in this study and whether it will largely affect the number of donors and data fitting?*
- 4. lines 38: The transmission from susceptible to infectious population.*

Response to reviewer: We realized that all comments 1,3, and 4 are concentrated on the transmission dynamics model and related parameterization. We fully agree with the reviewer's comments on the transmission model and concerns on the parameterization and fitting. All of these lead to the concerns about the projected cases used in the simulation of treatment dynamics. And we realized that most predictions on the COVID-19 outbreaks could be imprecise and proven wrong after a few weeks – but projection is not the major point of this study. The major goal of this study is to simulate the treatment dynamics with the implementation of the blood transfusion therapy, which can be fully decoupled from the transmission dynamics. The only ingredients we need from the transmission dynamics are: the inflow rate of newly identified cases, and estimated values of the symptom deterioration rate and the death rate. Therefore, we completely dropped the transmission model in our revised version, and replaced it by a baseline model that only focuses on the treatment dynamics of patients so as to fit the required parameters to the real data. To further avoid the uncertainties caused by projections, we utilized the real case count data by the end of year 2020 as the inflow rate of newly identified cases, and conducted all simulations to the same time point. We believe these modifications would not only address the concerns raised by the reviewer, but also make our results more robust (as it no longer depends on projections or how the transmission dynamics are modeled).